# Qualitative study of gesture annotation corpus: Challenges and perspectives

Mickaëlla Grondin-Verdon
Université de Lorraine, CNRS, Inria, LORIA
Nancy, France
Université Paul-Valéry Montpellier, CNRS, Praxiling
Montpellier, France
mickaella.grondin-verdon@loria.fr

Domitille Caillat
Université de Lorraine, CNRS, Inria, LORIA
F-54000, Nancy, France
Université Paul-Valéry Montpellier, CNRS, Praxiling
Montpellier, France
domitille.caillat@univ-montp3.fr

Slim Ouni
Université de Lorraine, CNRS, Inria, LORIA
F-54000, Nancy, France
slim.ouni@loria.fr

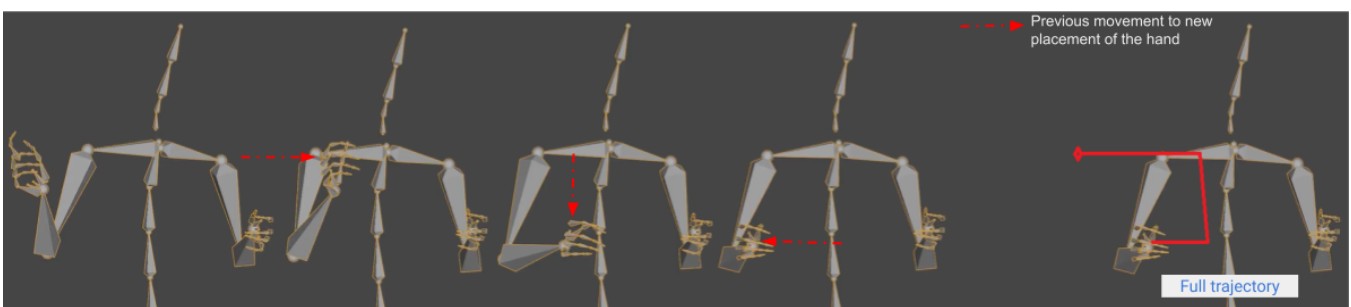

**Figure 1: Gesture example of a speaker in the BEAT corpus accompanying the word "drawing" with an iconic gesture**

## ABSTRACT

Effective data management and corpus enrichment are essential for advancing research methodologies in gesture studies. This paper critically examines the practices surrounding data management and corpora enrichment within a gesture dataset, focusing on qualitative analysis and methodological challenges. It identifies key issues in gesture annotation, including segmentation, labeling gestures, and lexical affiliates, revealing significant discrepancies and highlighting the complexities in interpretation. Despite these challenges, the inclusion of gesture dataset annotations marks progress in gesture research, offering opportunities for refining methodologies and enhancing data utilization. Strategies proposed aim to improve annotation practices, promote methodological transparency, and ensure the reliability of enriched corpora for nuanced analysis in gesture studies. This study contributes to advancing gesture research methodologies, emphasizing the importance of rigorous annotation protocols and fostering a standardized approach to enhance the utility and reliability of annotated datasets.

## CCS CONCEPTS

• **Applied computing** → **Annotation**; • **Computing methodologies** → **Lexical semantics**; • **General and reference** → *Reliability*; • **Human-centered computing** → *Human computer interaction (HCI)*.

## KEYWORDS

Gesture corpora, Enrichment, Annotation, Lexical affiliates, Gesture labelling

**ACM Reference Format:**
Mickaëlla Grondin-Verdon, Domitille Caillat, and Slim Ouni. 2024. Qualitative study of gesture annotation corpus: Challenges and perspectives. In *INTERNATIONAL CONFERENCE ON MULTIMODAL INTERACTION (ICMI Companion '24), November 4–8, 2024, San Jose, Costa Rica.* ACM, New York, NY, USA, 9 pages. https://doi.org/10.1145/3686215.3688820

## 1 INTRODUCTION

Human communication is a complex interplay of verbal and nonverbal elements, where gestures play a crucial role in conveying meaning beyond spoken words. Pioneering scholars, like Kendon, McNeill and more, have extensively explored the field of gesture studies, elucidating diverse concepts from gesture flow to functional description of gestures [11, 12, 21, 22]. Defined as any visible movement of a body part performed to communicate [5], gestures encompass a wide array of body movements, ranging from head to toe. These movements serve as vital components of human communication, coordinated with verbal discourse to convey meaning and intention.

Advancing the development of gesture generation models and enabling automated classification hinges critically on the availability of enriched gesture corpora. These corpora play a pivotal role in training models capable of performing tasks such as accurately simulating or classifying human gestures across diverse contexts [14]. Several datasets are currently employed for studying gestures and gesture generation. While the following examples provided are not exhaustive, they include several notable ones [2–4, 8, 9, 17–19, 26]. Among the datasets leveraged for these purposes, the TED Dataset stands out, drawn from TED conference videos and extensively employed in research focusing on the correlation between gestures and speech in conference settings [26]. Another significant resource is the Trinity Speech Gestures (TSG) dataset, encompassing TSG-I and TSG-II versions, which capture single-person motions discussing various topics and are instrumental for studying gesture production in natural speech environments [8, 9]. Additionally, the NOvice eXpert Interaction database (NoXi) provides insights into gesture usage during interactions between novices and experts in instructional contexts [4]. The Corpus of Interactional Data (CID) offers rich multimodal data capturing interactions in French, facilitating research on co-speech gestures and their communicative functions [2, 3]. The TalkingWithHands 16M dataset is notable for its extensive collection of multimodal data focused specifically on hand gestures in communication contexts [17]. The SaGA corpus consists of 25 dialogues between interlocutors, captured in both video and audio, where they engage in spatial communication tasks involving direction-giving and scene description [19]. Lastly, the Body-Expression-Audio-Text (BEAT) dataset [18] is notable for its very large scale and high-quality motion capture data, designed for gesture synthesis, cross-modality analysis, and emotional gesture recognition by providing extensive data. These datasets collectively enable investigations into gesture-speech dynamics and advancements, as in language sciences or in gesture generation modeling. However, despite their respective strengths and weaknesses, the majority of existing gesture corpora lack semantic enrichment, with the exceptions of the BEAT dataset, the SaGa corpus and the CID corpus. The CID corpus, however, lacks motion capture and suffers from poor video quality. In contrast, the SaGA dataset, while extensively annotated, is in German and also lacks motion capture. The BEAT corpus, on the other hand, has some annotation issues that will be analyze further. Consequently, most models do not leverage the features provided by such annotations because they are absent in these datasets, or hardly reusable. Clean and well-annotated data are crucial to enrich models with functional semantic elements that define gesture and its role in discourse. This approach could significantly enhance gesture generation outcomes. Moreover, it would be pivotal for developing enhanced automatic classification systems and class prediction, thereby facilitating semi-automatic annotation of corpora.

Semantic enrichment in the context of gesture studies involves various annotations, such as transcription of speech spanning from global utterances (what is said) to individual phonetic units and prosodic characteristics (how it is pronounced) or gesture annotations (what movements accompany the speech). Gesture segmentation is a critical process in gesture studies that involves breaking down continuous body movements into meaningful units for analysis. One prominent segmentation method follows Kendon's framework [11], which delineates various levels of units. At the highest level, Kendon defines the "gesture unit" as the broadest observable segmentation of gestural flow. It is characterized by the transition from one resting position to another, marked by a relaxation of the involved body parts. During this defined interval, the individual performing the gesture can execute one or multiple consecutive gestures. Within Kendon's framework, a more detailed segmentation approach involves identifying "gesture phrases", which constitute individual gestures within gesture units. This segmentation includes distinct phases: preparation, stroke, retraction, and hold, with the stroke phase being mandatory. It is possible to annotate only the stroke, as it represents the minimal unit of movement that preserves the semantic integrity of the gesture, encapsulating its core movement and conveying its essential meaning [11]. These gesture segments are then labeled according to a specific typology chosen based on the study's objectives, enabling a more detailed and contextual analysis of gestures in discourse. Gestures can be categorized and studied either by their form or their function within discourse, with much of the research focusing on manual gestures using the McNeill typology [21] as a primary descriptor. McNeill's functional classification categorizes gestures into distinct types: beats, deictic, iconic, and metaphoric, each serving unique communicative functions. Beat gestures serve as emphasizing movements synchronized with the rhythmic characteristics of speech. Deictic gestures, such as pointing, refer to specific referential elements in speech, such as objects, persons, time or locations dependant on the situation of enunciation. Iconic gestures visually represent actions or concepts, aiding in concrete representation based on the spatio-graphic, pictographic, or kinematic characteristics of an element in discourse. Metaphoric gestures employ movements symbolically to convey complex concepts, enriching the spoken discourse through visual expression. Annotations can also include labeling the lexical affiliate [13, 25] or prosodic affiliate of a gesture. A lexical affiliate refers to a specific word, expression or phrase in spoken language that a gesture is closely associated with, providing a direct link between the gesture and its semantic content. For instance, a pointing gesture may be directly linked to the word "there" in a sentence to reference somewhere in the environment. On the other hand, a prosodic affiliate involves the relationship between a gesture and the prosodic features of speech, usually linking beat gestures to specific syllables. This affiliation highlights how gestures can be synchronized with the acoustic patterns of speech to enhance communicative clarity and emphasis.

These approaches enable a more nuanced and structured analysis of gestures, thereby facilitating their interpretation and use in automated gesture generation and classification tasks. However, only a few corpora include these types of annotations, posing a significant limitation in gesture research. Moreover, the lack of minimal standardization across corpora can lead to inconsistencies and omissions in crucial elements for their exploitation and analysis, particularly in an open science context. Creating a gesture corpus is a costly endeavor, both financially and in terms of time investment. Collecting and annotating data, such as detailed transcriptions and gesture segmentation analyses, require considerable effort. Indeed, these processes require the participation of qualified experts, along

with the use of specialized tools. Resorting to non expert annotators cannot be done without the implementation of trusted protocols to ensure accurate and reliable results. This increases the complexity and costs associated with constructing a high-quality corpus and likely explains why there are relatively few such corpora available.

During the recent GENEA Challenge 2023, it was found that none of the methods tested for generating speech-accompanying gestures could realistically replicate human gestural behavior [15]. This underscores that the challenge of accurately simulating human gestures remains unresolved. A key factor contributing to this limitation is the use of unlabeled datasets for training, where gestures often lack contextual relevance to discourse and production contexts. As a result, the quality of training data in corpora becomes a critical issue, highlighting the importance of naturalness and richness in observed gestures, as well as the accuracy of associated annotations. The BEAT corpus [18], with its fully annotated and multimodal data, stands out as the only corpus currently offering semantic elements for gesture analysis. This corpus claims to offer detailed annotations of gesture semantic aspects, labeling them based on McNeill's classification and incorporating labels of lexical affiliate of the gesture when applicable. Notably, due to the quality of its motion capture data, BEAT is increasingly recognized as a valuable resource for gesture generation research. BEAT was proposed to participants of the GENEA Challenge 2023 [15], underscoring its significance in the field of gesture generation research. This article focuses on the challenges and methodologies of data management and corpus enrichment within gesture research, using the BEAT dataset as a case study. Our objective is to conduct a qualitative analysis of the annotations provided in the BEAT corpus, a task that is uncommon and has not been previously undertaken. Given the scarcity of such data and the challenges involved in creating them, our aim is to verify that these annotations are reliable and can be reused with confidence. This study delves into the qualitative examination of a gesture annotation corpus, exploring both the challenges it poses and the potential for refining methodologies to improve its efficacy.

## 2 ANALYSIS OF THE BEAT DATASET

### 2.1 Description of dataset

The Body Expression Audio Text (BEAT) [18] corpus is a large-scale multi-modal and muti-language dataset. The corpus contains 76 hours of recordings in various languages: 60 hours in English, 12 hours in Chinese, and 2 hours each in Spanish and Japanese. It includes high-quality data: motion capture for body, hands and face, as well as audio. The motion capture was realised with 16 synchronized cameras operating at 120 Hz and participants wore Vicon's suits with 77 refletive markers. Arkit and depth camera were used for the facial motion system, to extract 52 blendshape weights at 60 Hz, desgined on Facial action coding system, FACS [7]. Additionally, it claims to provide various annotations: text transcriptions at both the word level using an in-house-built ASR model and the phoneme level using the Montreal Forced Aligner, MFA [20], which relies on Kaldi [23], emotion annotations at the recording level, and semantic annotations at the gesture level.

BEAT is evenly split between conversation sessions and self-talk sessions, with sequences lasting 10 minutes and 1 minute,

respectively. Topics were chosen from a set of 20 predefined topics, encompassing 33% debate topics and 67% description topics. Conversation sessions captured neutral conversations naturally, without prompting. Self-talk sessions included 120 recordings, where actors delivered scripted responses to questions on everyday conversation topics, playing out answers that had been previously collected. Out of the 120 questions, 64 were related to neutral emotions, while the remaining questions were divided equally among seven other emotional categories. Speakers were asked to discuss the same content using their own personalized gestures and to read answers in self-talk sections. They would watch 2-10 minutes of emotionally stimulating videos corresponding to different emotions before speaking with the specific emotion. For BEAT's semantic annotations, annotators reviewed videos with synchronized audio and gestures to perform frame-level annotations. From an initial pool of 600 annotators recruited from Amazon Mechanical Turk (AMT), 118 were selected after successfully completing a small test dataset for qualification. After a video demonstration and introductory summary, annotators assessed semantic relevance using a scale ranging from 0 to 10 assigning a single score at a time for each gesture. These scores were associated with different types of gestures (Table 1) and were introduced by the creators of BEAT with associated common words during annotation task, as "here", "that" or "this" for deictics, "driving", "run" or "one" for iconics and "future", "past" or "direct" for metaphorics.

### 2.2 Qualitative analysis methodology

Our primary focus was on conducting qualitative analysis of BEAT gesture annotations using traditional data management and specialized tools for annotations purposes. For the comparative analysis of annotations, we performed our own gesture segmentation, gesture labeling and affiliate labeling on an extract from the BEAT dataset, utilizing reconstructed movements derived from motion data and synchronized audio provided by BEAT. The annotations presented in this article serve as illustrative examples, and do not encompass the entirety of our work. While only a subset of our annotations is shown here, this example is one of many from our ongoing study conducted by our expert annotators. Two expert annotators in gesture studies within the linguistic field performed the annotation, with a joint review process for segmentation, affiliation, and labeling to ensure consistency and accuracy. When labeling gestures in this context, we follow McNeill's categorization [21, 22] into metaphoric, deictic, iconic, and beats gestures so rather than directly using the scores like in BEAT. We employed a multidimensional annotation approach, allowing for the indication of two labels when relevant. To compare our results with those of the BEAT annotators, we translated the score annotated into their corresponding labels according to McNeill's framework. As part of proper gesture annotation, we expect segmentation to include at least the stroke phase described by Kendon's framework [11, 12]. Intended to feed a gesture generation model, our annotations were limited for each gesture to identifying this key phase only, assuming that the model will be able to determine the movements (preparation, retraction or other phase if needed) required to achieve these positions which carry meaning. Additionally, it is crucial that each

**Table 1: Annotations instructions in BEAT**

| Annotation | Label |
|---|---|
| 0 | No gesture |
| 2-4 | Low to high quality deictic gestures |
| 5-7 | Low to high quality iconic gestures |
| 8-10 | Low to high quality metaphoric gestures |
| Habit | Gestures not related to speech. |

**Table 2: Annotation categories analysis in BEAT**

| Category | Count | Count% | Duration (h:m:s) | Time% |
|---|---|---|---|---|
| beat | 20914 | 51.25% | 54:42:2.775 | 88.27% |
| iconic | 8162 | 20.00% | 2:51:13.471 | 4.61% |
| metaphoric | 6933 | 16.99% | 2:21:10.560 | 3.80% |
| deictic | 4468 | 10.95% | 1:32:30.669 | 2.49% |
| habit | 255 | 0.62% | 0:12:53.661 | 0.35% |
| nogesture | 75 | 0.18% | 0:17:50.349 | 0.48% |
| **Total** | **40807** | **100%** | **61h 1m 1.485s** | **100%** |

segmentation pertains to a single gesture, avoiding the inclusion of multiple gestures within one segmentation.

For the analysis of affiliate errors and duration analysis, we utilized the entire BEAT dataset to evaluate word recognition accuracy and timestamps duration. Specifically, we employed spaCy [10], a robust natural language processing library, to determine whether each word in the annotations was recognized as a valid English word by the language model. The comparative analysis between our re-annotated affiliates and those in the BEAT dataset serves as an illustrative example in this article, highlighting how discrepancies in lexical affiliate annotations can reveal broader issues in gesture labeling accuracy. Specifically, inconsistencies in affiliate annotations can reflect underlying challenges in the overall quality of gesture segmentation and labeling within the dataset. The primary focus of our study on affiliates is the analysis of orthographic errors within the entire BEAT dataset, as these errors can significantly hinder semantic analysis by misrepresenting the intended meaning of affiliates.

## 2.3 Annotation categories in BEAT

The English-speaking part includes 34 hours of recordings featuring 10 native English speakers from the US, UK, and Australia, along with 26 hours from 20 fluent English speakers from other countries. This article focuses extensively on the English-speaking component of the BEAT dataset. There are a total of 40807 annotations (Table 2). Beat gestures emerge as the predominant category, comprising 51.25% of all annotations, totaling 54 hours, 42 minutes, and 2.775 seconds of recorded activity, which accounts for 88.28% of the total dataset duration. Following closely are iconic gestures, accounting for 20% of the annotations, corresponding to 2 hours, 51 minutes, and 13.471 seconds of duration, or 4.61% of the dataset duration. Metaphoric gestures constitute 17% of the annotations, reflecting a duration of 2 hours, 21 minutes, and 10.560 seconds, contributing 3.80% to the dataset. Deictic gestures, with 11% of the annotations, encompass 1 hour, 32 minutes, and 30.669 seconds of duration, representing 2.49% of the total dataset duration. Habit gestures are less

frequent, comprising 0.63% of annotations, with a total duration of 12 minutes and 53.661 seconds. Finally, no gesture instances are the least frequent, accounting for less than 0.2% of annotations, with a duration of 17 minutes and 50.349 seconds. Based on the comprehensive annotation counting and total duration analysis, it can be concluded that participants in the BEAT dataset predominantly engage in gesturing throughout their recorded sessions. The data reveal that beat gestures, comprising the majority of annotations and total duration, indicate that participants spend the majority of their time gesturing, with minimal to no periods of rest or pauses which is unusual. The activity is predominantly characterized by beat gestures, indicating continuous engagement and minimal idle moments throughout the recorded sessions.

## 2.4 Annotation analysis

*2.4.1 Annotation segmentation and gesture labeling.* Figure 2 shows a representation of part of our annotation analysis of the BEAT dataset, specifically from speaker 1 (Wayne), file 0_53, spanning from 00:13.500 to 00:18.500. It includes annotations from the dataset compared to our proposed annotations for this segment. During this interval, the speaker, Wayne, says "everything is based on drawing I think drawing is the best of almost any form of art". In the BEAT dataset, Wayne's gestures were annotated as continuous gestures, including three beats, one iconic and one metaphoric. It should be noted that the first and last beats begin before 00:13.500 and end after 00:18.500. The iconic gesture was annotated with the lexical affiliate "drawing" and the metaphoric gesture with "best". During our analysis of this segment, we observed significant differences in annotation compared to BEAT annotations. We annotated shorter beats, an iconic gesture and a metaphoric gesture, and not placed at the same specific moments. Additionally, lexical affiliations have been annotated differently based on our interpretation. In our proposed annotations, the beats are affiliated with syllables ("based", "best", and "draw" from "drawing"), whereas in BEAT, no affiliates were annotated for beat gestures (see Table 3 for detailed number of affiliates per category). The iconic labeling we propose appears similar to BEAT's, though the timestamps differ —annotated from 00:15.504 to 00:16.436 in our case compared to 00:14.202 to 00:15.056 in BEAT. The lexical affiliate is similar, but due to the time position and contextual speech, it's unclear if their affiliate is for the first or second instance of "drawing" verbalized by Wayne. In our annotation, the lexical affiliate "drawing" pertains to the second mention of it in Wayne's speech. Figure 1 illustrates our segmentation and interpretation, which differs from the one proposed in the BEAT corpus for the iconic gesture in the segment from 00:13.500 to 00:18.50. The BEAT corpus suggests a broader segmentation than ours and interprets it as a beat gesture. Based on contextual analysis, including the participant's utterance, a square-shaped gesture could be interpreted as iconic, particularly with "drawing" as its lexical affiliate. This interpretation suggests that the gesture represents the form of the drawing itself rather than emphasizing a beat or rhythm in speech, thereby justifying its classification as iconic rather than beat. The metaphorical gesture follow a similar analysis with different timestamps observed —annotated from 00:17.537 to 00:18.144 in our case compared to 00:16.712 to 00:17.712 in BEAT. In our segmentation, Wayne initially holds both hands close in front

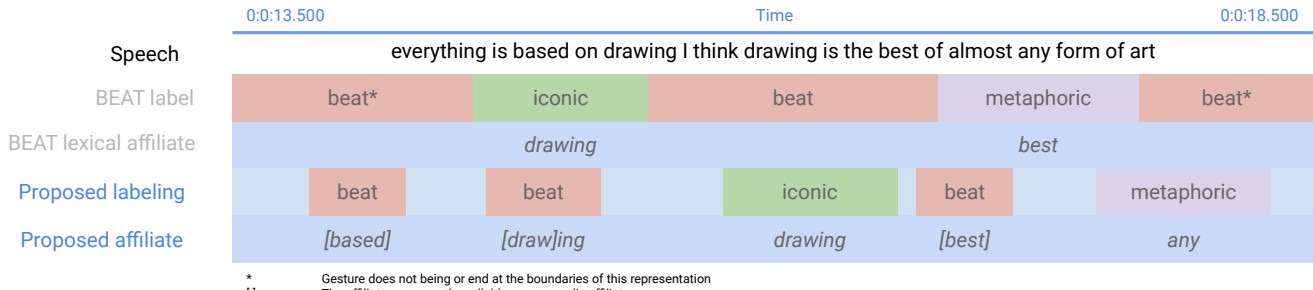

**Figure 2: Analysis of pre-existing gesture annotations in the BEAT dataset and our proposed annotations for this segment**

**Table 3: Summary of categories with affiliates, mean and median of the number of words annotated as lexical affiliates and number of errors**

| Category | Count | Mean | Median | Errors |
|---|---|---|---|---|
| beat | 0 | - | - | - |
| iconic | 8145 | 3.775 | 2 | 69 (0.85%) |
| metaphoric | 6922 | 3.685 | 2 | 42 (0.61%) |
| deictic | 4468 | 3.677 | 2 | 47 (1.05%) |
| habit | 255 | 2.196 | 2 | 3 (1.18%) |
| nogesture | 0 | - | - | - |
| **Global** | **19790** | **2.329** | **1** | **161 (0.81%)** |

**Table 4: Gesture durations analysis (seconds) in BEAT**

| Category | Mean | Median | Std_Dev | Min | Max |
|---|---|---|---|---|---|
| beat | 9.42 | 4.34 | 17.54 | 0.10 | 643.00 |
| iconic | 1.26 | 1.00 | 0.59 | 0.21 | 9.60 |
| metaphoric | 1.22 | 1.00 | 0.51 | 0.29 | 8.73 |
| deictic | 1.24 | 1.00 | 0.81 | 0.16 | 14.83 |
| habit | 3.03 | 1.78 | 3.62 | 0.59 | 26.82 |
| nogesture | 14.27 | 10.90 | 10.76 | 3.58 | 56.19 |
| **Global** | **5.47** | **1.479** | **13.22** | **0.10** | **643.00** |

Every habit gesture had at least one word as an affiliate, with a mean of 2.196 and a median of 2 words as affiliates per gesture, and an error rate of 1.18% (3 errors detected). As an example, in file 1_1 of speaker Zhang, a metaphoric gesture has the affiliate "enbironnment." Since "enbironnment" is not a valid word, this is counted as an error due to an orthographic mistake. Another cause of errors in the affiliates in BEAT is missing spaces, as seen in file 0_11 of speaker Li. In this instance, the affiliate of the iconic gesture is "fashion magazines and the inspirationbooks," with a missing space between "inspiration" and "books," making it a non-word. Additionally, in file 0_6 of speaker Carla, a deictic gesture has the affiliate "I.didn't". The inclusion of the period instead of a space creates an error, resulting in an invalid token in SpaCy. Overall, the majority of words are recognized by spaCy, but iconic, metaphoric, and deictic gestures have some instances of unrecognized words. Not recognized words might include proper nouns that are not recognized by spaCy but are used as affiliates. However, these findings highlight the importance of regular human revision to ensure the quality and accuracy of annotations corpora. This is particularly crucial when such data is used for analyses and research that require precise textual data, as errors can potentially compromising the interpretation of the data.

## 2.5 Annotation duration analysis

The analysis of gesture duration in the BEAT dataset provides insights into the distinctive characteristics of different gesture segments, as summarized in Table 4. We observed that the mean duration of a gesture in BEAT is 5.47 seconds, while the median duration is 1.5 seconds. This disparity between the median and mean durations suggests notable variability in gesture durations across the dataset. The standard deviation (Std_Dev) of 13.22 seconds further underscores this variability, indicating a wide spread of durations. The analysis of gesture durations across different categories reveals

of their body. Suddenly, they swiftly raise their right hand straight up into the air. We interpreted the lexical affiliate differently, identifying "any" as the lexical affiliate for the metaphoric gesture (which represents the extent or range of something abstract), whereas in BEAT "best" was annotated as such.

*2.4.2 Affiliates annotations.* The analysis of affiliates in the BEAT dataset reveals distinct error rates across different gesture categories, as summarized in Table 3. Beat gestures and "no gesture" labels had no affiliates annotated, resulting in their exclusion from this analysis. However, it's important to note that beats can indeed have affiliates, despite their exclusion from this analysis due to the lack of annotated affiliates. As affiliates can consist of one or more words, a single error is counted if at least one word in the affiliate is not recognized. Therefore, if two words in the affiliate are incorrect, the entire affiliate is considered incorrect due to the one-error rule. The mean and median values presented in Table 3 are calculated based on the number of words per gesture affiliate. The overall results presented in the table indicate that out of a total of 19790 affiliates analyzed in the BEAT corpus, 161 errors were detected by spaCy, corresponding to a global error rate of 0.81%. This metric encompasses all gesture categories included in the analysis: iconic, metaphoric, deictic, and habit. Iconic gesture affiliates, with 8145 instances, had a mean of 3.775 and a median of 2 words as affiliates per gesture, with an error rate of approximately 0.85% (69 errors detected by spaCy). Metaphoric gestures, numbering 6922 affiliates, had a mean of 3.685 and a median of 2 words per affiliates, with an error rate of about 0.61% (42 errors). Deictic gesture affiliates, observed in 4468 instances, had a mean of 3.677 and a median of 2 words as affiliate per gesture, with an error rate of 1.05% (47 errors).

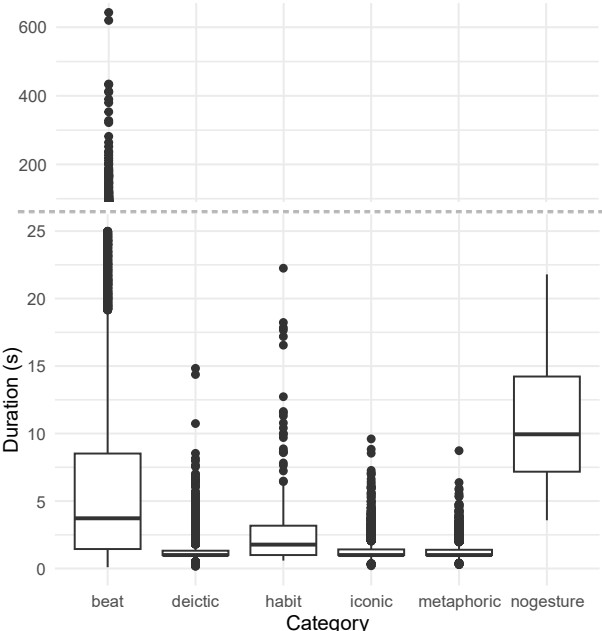

**Figure 3: Visual representation duration values per categories in BEAT**

the origins of this variability, highlighting distinct patterns and characteristics within each category. Gestures categorized as beat exhibit a relatively high average duration of 9.42 seconds, with a median of 4.34 seconds, with a standard deviation of 17.54 seconds, indicating a high variability in their durations. The range spans from 0.10 to 643.00 seconds, highlighting substantial outliers with tendencies of long durations as the mean is superior to the meadian. In contrast, the iconic, metaphoric, and deictic gestures are shorter in duration, averaging around 1.2 seconds with low variability (standard deviations less than 1 second). Specifically, iconic gestures have an average duration of 1.26 seconds, metaphoric gestures average 1.22 seconds, and deictic gestures average 1.24 seconds, each with median durations around 1.00 second. The habit category shows gestures averaging 3.03 seconds, indicating moderate variability (standard deviation of 3.62 seconds) and ranging from 0.59 to 26.82 seconds. Notably, segments categorized as nogesture have long average durations of 14.27 seconds, with considerable variability (standard deviation of 10.76 seconds) and ranging from 3.58 to 56.19 seconds, reflecting extended periods without gestures interspersed within the corpus. This suggests that there are naturally occurring intervals where the speaker does not perform gestures, hence being classified as nogesture. However, due to the relatively low frequency of nogesture annotations, such moments are quite rare. The "beat" and nogesture categories exhibit significantly higher mean and median durations compared to other categories, suggesting either longer gestures or longer segments without gestures. The "beat" category shows substantial variability, reflected in its large standard deviation and range (min-max), possibly due to the presence of exceptionally long gestures. In contrast, iconic, metaphoric, and deictic categories display relatively similar durations, with means and medians around 1 second and low standard deviations,

**Table 5: Non-extreme outliers (Out.) and extreme outliers (Ext.) analysis by category (Cat.) in the BEAT dataset.**

| Cat. | Out. count | Ext. count | Total |
|---|---|---|---|
| beat | 1005 (4.8%) | 797 (3.81%) | 1802 (8.61%) |
| iconic | 389 (4.8%) | 215 (2.63%) | 604(7.4%) |
| metaphoric | 338 (4.9%) | 159 (2.3%) | 487 (7.2%) |
| deictic | 389 (8.70%) | 290 (6.5%) | 679 (15.2%) |
| habit | 9 (3.53%) | 15 (5.88%) | 24 (9.41%) |
| nogesture | 3 (4%) | 4 (4%) | 6 (8%) |
| **Global** | **2133 (5.23%)** | **1479 (3.62%)** | **3612 (8.85%)** |

indicating relatively short and consistent gestures within these categories. As depicted in figure 3, it is evident that many durations appear to be outliers across all categories, as indicated by numerous data points lying far outside the interquartile range represented by the boxplots. Especially noticeable are numerous data points far from the mean duration of gestures in the beat category. This observation contrasts sharply with the distributions of durations in the other annotated categories. Specifically, categories such as iconic, metaphoric, deictic, habit, and nogesture show data points that are relatively closer to the median and interquartile ranges, suggesting more clustered and less variable durations within these categories compared to beat gestures. This disparity underscores the unique distributional characteristics of gesture durations within the beat category, where outliers significantly influence the overall distribution pattern observed in the corpus analysis.

*2.5.1 Duration outliers analysis.* An outlier is an observation that falls outside the expected range of normal values within a dataset, possibly due to measurement errors, genuine but rare extreme values, or other unusual causes. An extreme outlier is an observation that lies even farther from the mean or quartiles compared to other outliers. In the analysis of outliers in the BEAT dataset (Table 5), we identified a total of 3612 durations classified as outliers, which represents 8.85% of all duration values in the dataset (40807 annotations). Among the outliers identified, 1479 durations were classified as extreme outliers, comprising 3.62% of the entire dataset. This subset of durations significantly deviates from the typical distribution observed in the dataset, highlighting potential anomalies or exceptional cases in gesture durations within specific contexts or categories. The figure 4 provides a clear overview of how outliers

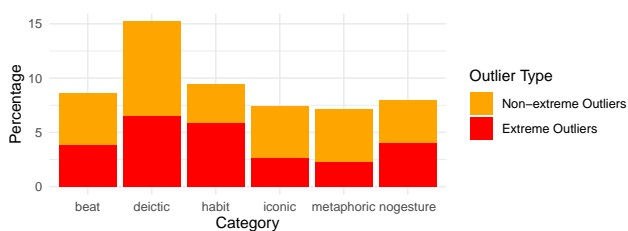

**Figure 4: Visual representation of non-extreme and extreme outlier durations by category in the BEAT dataset.**

are distributed among gesture categories, highlighting variations in outlier prevalence across different types of gestures. In the beat

category, there are 1,802 outliers, accounting for 8.61% of all beat gestures, with 797 of these being extreme outliers, representing 3.81% of all beat gesture durations. In the deictic category, 679 outliers were observed, among which 290 are extreme outliers, making up 6.49% of all deictic gestures. Gestures categorized as habit show 24 outliers, comprising 9.41% of this category, with 15 of them being extreme outliers, accounting for 5.88% of all habit gestures. In the iconic category, there are 604 outliers, equivalent to 7.40% of this category, with 215 being extreme outliers, representing 2.63% of all iconic gestures and 35.6% of its outliers. The metaphoric category exhibits 497 outliers, making up 7.17% of this category, with 159 being extreme outliers, accounting for 2.3% of all metaphoric gestures. Finally, in the nogesture category, 6 gesture segments are identified as outliers, constituting 8% of this category, with 3 being extreme outliers, representing 4% of all nogesture segments. Most of these data points exhibit durations significantly longer than the database average, indicating an asymmetric distribution of gesture durations, with a notable proportion of outliers being much higher than normal. Additionally, there is a significant portion of outliers in each category that are classified as extreme outliers.

## 3  DISCUSSION

Annotation errors are prevalent during corpus creation due to the intricate nature of the task. Mitigating these errors is essential to uphold data quality and reliability. Through the analysis of gesture segmentation, duration, labeling, and affiliates annotation, significant issues and insights concerning the BEAT dataset annotations come to light.

Our study reveals significant divergences in annotation compared to the BEAT dataset. Various types of errors were observed, including segmentation errors where segments fail to isolate gestures —at least the gesture phases— resulting in subsequent gestures being misrepresented in terms of quantity and making it difficult to reliably associate them with the discourse. Consequently, labeling errors may also occur, leading to different interpretations. The interpretation of gestures, such as the iconic gesture representing "drawing" in Wayne's speech, highlights challenges in contextual analysis. Our segmentation argued for an iconic classification based on the gesture's representation of the drawing's form, diverging from BEAT's broader segmentation as a beat gesture. This difference underscores the complexity of gesture interpretation and the influence of contextual factors on annotation decisions. The discrepancies observed underscore the necessity for rigorous methodologies in gesture annotation. Variations in annotation criteria, timing, and lexical affiliation can lead to differing interpretations and classifications, affecting the reliability and reproducibility of research findings. Addressing these methodological challenges is crucial for advancing the field of gesture studies and ensuring robust and comparable datasets. This suggests variability and subjectivity in gesture annotation methodologies, impacting the consistency and comparability of datasets in gesture research.

The analysis of affiliate annotations in the BEAT dataset reveals varying error rates across different gesture categories, underscoring specific challenges such as unrecognized words due to spelling errors or missing spaces between words. These errors, though minimal in percentage terms (ranging from 0.61% to 1.05% across categories), can significantly impact data interpretation. Precise lexical affiliations are crucial for understanding gesture meanings within contextual speech. While the overall error rate is low (0.81% globally), addressing all possible errors in data management is imperative to maintain the integrity and accuracy of gesture annotations. This diligence is essential given that even minor discrepancies can potentially distort the interpretation of gestures in discourse analysis and related research contexts.

The analysis of gesture durations in the BEAT dataset offers valuable insights into the variability and distribution patterns within the corpus. The observed difference between mean and median durations across categories emphasizes the existence of outliers and underscores the inherent variability in gesture durations. Gestures categorized as beat exhibit significantly longer average durations compared to other categories, with a notable standard deviation indicating a wide range from very short to exceptionally long gestures. Excessively long gestures have the potential to distort the analysis of gestural dynamics, particularly if they do not accurately represent the actual gestures performed during recordings. Moreover, it is particularly unusual that there is very little rest time between gestures and that annotations are continuous, as if speakers are continuously gesturing, which contrasts with observations in the motion data visualization. The prevalence of outliers, especially within the beat category, underscores the critical importance of carefully assessing and interpreting gesture durations during corpus annotations. These outliers can greatly influence the overall distribution and interpretation of gesture behaviors within specific contexts or communicative scenarios.

Annotation errors are widespread, highlighting significant challenges in the verification process and emphasizing the critical importance of rigorous data management and annotation practices to ensure the reliability and validity of gesture research datasets. Within BEAT, the annotation process suffers from a lack of comprehensive documentation and methodology clarity, notably lacking a detailed annotation coding manual. Introducing a systematic documentation is essential to clarify research protocols, facilitate precise replication of processes, conventions, guidelines, and criteria used by the corpus creators. This manual would provide annotators with a clear reference, standardizing annotation methods, mitigating interpretation errors, and ensuring consistency across different project phases. To enhance methodological transparency and reproducibility, it is crucial to standardize the inclusion of such documentation as a requirement for publishing enriched corpora, as done by the authors of the SaGA corpus ([1]). Similar documentation practices exist in language sciences (e.g., [6, 16, 24]), but the complexity of such manuals should not hinder the understanding of the protocol. Therefore, it is essential to strike a balance between detail and accessibility to ensure that the guidelines are both comprehensive and user-friendly.

In addition to this document, standardized measures should be implemented to ensure best practices. Firstly, a synchronization phase among annotators is essential to achieve consistent annotation practices. Providing the annotation manual alone to annotators is not sufficient to ensure this consistency. Such training familiarizes annotators with the annotation task using sample datasets,

reduces ambiguity, and enhances consistency in annotation practices. Establishing a synchronization phase where annotators review and discuss their annotations is critical. Group discussions facilitate shared understanding and align annotators' interpretations with the annotation manual. While group consensus can help reduce discrepancies, inconsistencies among annotators may lead to variability in segmentation, potentially accepting errors as the prevailing choice, thereby falsely perceived as objective. Assessing the quality of work by inexperienced annotators lacking training is particularly challenging due to the precision required. Therefore, establishing robust training and synchronization protocols is crucial to uphold the quality and reliability of annotations. Additionally, implementing community research protocols and baseline criteria for gesture annotations (e.g., anatomical landmarks for segmentation, mandatory phases, and complexity of documentation) is essential. Secondly, implementing robust quality assurance mechanisms throughout the annotation process is essential. This involves incorporating certainty scores into gesture annotations (e.g., from 0 to 5, from "no certainty at all" to "absolute certainty"), or regular reviews of annotated data by experienced supervisors or validators to detect errors, inconsistencies, or deviations from annotation guidelines. Certainty scores quantify confidence in the annotation of each gesture, allowing for measurement of the reliability of gesture classifications and identification of areas of uncertainty in gesture interpretation. Establishing feedback loops and conducting manual revisions, especially when using automatic tools (e.g., for transcription, affiliate selection, or segmentation), are crucial for correcting errors and enhancing annotation accuracy. This ensures that annotations remain high-quality and suitable for rigorous gesture analysis in future research.

Enriching a corpus is a crucial undertaking that demands significant time and financial investment. However, striking a balance between achieving high quality and expanding quantity presents inherent challenges. Expert annotators, renowned for their rigorous methodologies, are often in short supply, expensive, and not always accessible. To address these challenges, integrating inexperienced annotators can offer a practical solution. In the context of BEAT, this approach has notably augmented the volume of supplementary data available, which holds substantial value within the gesture research community. This expanded demand extends beyond basic recordings to encompass additional layers of annotation that enrich the analysis of gestures. A notable advancement in BEAT has been the incorporation of lexical affiliates, enhancing the depth and contextual understanding of the data —a feature that sets it apart from other gesture corpora. Despite the increased quantity of annotated data, challenges persist, particularly when conducting global recruitments through online platforms. This necessitates meticulous management to uphold data integrity when working with inexperienced annotators. It is crucial to limit the complexity of tasks assigned to inexperienced annotators to maintain accuracy and consistency in data annotation. Their lack of nuanced understanding and methodological expertise compared to experts can pose difficulties in handling highly complex tasks requiring precision and consistency. Therefore, researchers should tailor annotation tasks to match the skills and capacities of inexperienced annotators. This involves clearly structuring tasks, ensuring they are feasible, and aligning them with available resources.

In summary, effective management of tasks and expectations is essential when supervising inexperienced annotators. For inexperienced annotators, it is crucial to assign simpler tasks (e.g., basic movement descriptions), ensure proper segmentation, and distribute responsibilities effectively. Simplifying tasks allows them to grasp annotation concepts more easily and reduces the likelihood of errors. Clear segmentation guidelines help maintain consistency and accuracy in annotation practices, ensuring that each gesture or unit of analysis receives appropriate attention. Effective distribution of tasks among inexperienced annotators also spreads the workload evenly, optimizing their learning experience and overall annotation efficiency. This structured approach not only enhances the quality of annotations but also fosters the growth of annotators' skills over time. By nurturing their development, it contributes to the overall success of corpus enrichment initiatives while ensuring the sustained quality and reliability of the annotations produced.

## 4 CONCLUSION

While the analysis reveals major annotation errors and issues in BEAT, questioning the reliability of segmentation and labeling, it also underscores the progress achieved through the inclusion of these data. This case study could benefit from a thorough examination of the corpus and a deeper exploration of the annotations. Addressing these challenges is crucial for advancing gesture studies and creating robust, comparable datasets. Learning from these errors and refining practices will enhance the quality and reliability of annotated corpora and advance gesture analysis. This collaborative effort promotes a shared understanding of criteria, improves inter-rater reliability, and standardizes gesture annotation across the corpus. Reference annotations on natural movements set clear objectives for assessing gesture richness and provide benchmarks for evaluating gesture synthesis systems, ensuring they produce precise gestures with similar richness and naturalness as human behavior. These efforts offer objective evaluations that complement traditional perceptual assessments. Strengthening dataset validity and reliability is key for deeper insights and more impactful future research.

Improving the rigor and accuracy of gesture annotations to better capture the natural and communicative aspects of human behavior can significantly enhance the utility of enriched corpora. Capturing these nuances allows researchers to use these datasets more effectively in various applications. Future gesture research must focus on studying corpus enrichment practices. This involves refining annotation methods and validating data to better understand gestures and advance fields like human-computer interaction and embodied interactional agents. Researchers face challenges in generating richer and more realistic co-speech gestures, primarily due to limited annotated corpora. Annotation is key to overcoming this challenge. Without ample data, deep learning models struggle to generate complex gestures without prior knowledge. Fine-grained annotations within gesture corpora can effectively represent this knowledge. Detailed annotations give neural networks the structure and context to better replicate human gestures. Consequently, the advancement of gesture generation models is intrinsically tied to the development and use of meticulously annotated datasets, highlighting the critical importance of annotation in this domain.

# ACKNOWLEDGMENTS

This project has received financial support from the CNRS through the MITI interdisciplinary programs through its exploratory research program.

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
