# OpenReview forum: "Qualitative Study of Gesture Annotation Corpus: Challenges and Perspectives"
_ACM.org/ICMI/2024/Workshop/GENEA — GENEA Workshop 2024_

### Official Review · Reviewer_rK81 · 2024-07-17
**enriching gesture annotation should contribute the field**

**Rating:** 6
**Confidence:** 5

**Review:**

#### SUMMARY
The paper proposed to use conventional annotation method and annotated English-speaking part (54 hours) of BEAT dataset.
118 participants recruited from Amazon Mechanical Turk took part in this annotation.
The annotation is performed aware of gesture stroke phase and more precise lexical affilate. The detailed analysis of annotated results is provided.

#### STRENGTHS
- Updating/Enriching the gesture annotation of co-speech gesture dataset is very basics of data-driven approach, and it should contribute to the field.
- The paper introduce annotation method in humna-human gesture analysis research into data-driven co-speech gesture generation field.
- The paper is well-organized, the technical descriptions are well-written, and the experiments are reproducible.

#### WEAKNESSES
- (minor) Introduction needs a section number.
- McNeil's gesture classification is not categorical but dimensional. For example, beat and iconic emerges at the same time. I believe the proposed annotation can not put multiple gesture categories.
- The annotation is aware of gesture phase (rest, preparation, pre-stroke-hold, stroke, hold, ...), but they are not annotated. these information can be utilized in co-speech gesture generation.

**Nominate For A Reproducibility Award:**

no

---

### Official Review · Reviewer_AD8U · 2024-07-29
**An initial study of an important problem**

**Rating:** 7
**Confidence:** 5

**Review:**

SUMMARY

This paper is an initial study of the reliability of gesture annotations in the BEAT dataset, containing three analyses:
1. A 5-second segment is compared to a re-annotation performed by the authors, hinting at potentially systematic inaccuracies in gesture segmentation and affiliate assignment.
2. A word recognition test is performed on all lexical affiliates, identifying 0.81% of affiliates as likely erroneous.
3. An analysis of gesture durations reveals a relatively high ratio outlier durations across all gesture categories, with a significant portion being extreme outliers.

Based on these results, the authors discuss the issues of the gesture-annotation methodology in BEAT, and propose several solutions for a more rigorous annotation.

STRENGTHS

* The paper addresses an important and understudied aspect of gesture generation.
* The paper is well-written with detailed background and discussion sections.
* The experiments reveal various shortcomings of the gesture annotations in one of the most popular motion-capture datasets in gesture generations.
* A detailed discussion on gesture-annotation methodology that will help the community move forward.

WEAKNESSES

* Concrete examples (i.e., a mock-up) for some of the methodological suggestions - e.g., how to separate annotation tasks between expert and non-expert annotators - would be a valuable addition to the paper.
* The SaGA dataset [1] is missing from the related work. It is a non-mocap dataset with very detailed gesture annotations and a public annotation manual [2].


REFERENCES

[1] Lücking, Andy, Bergmann, Kirsten, Hahn, Florian, Kopp, Stefan, and Rieser, Hannes. 2010. “The Bielefeld Speech and Gesture Alignment Corpus (SaGA)”. In LREC 2010 Workshop: Multimodal Corpora–Advances in Capturing, Coding and Analyzing Multimodality, ed. M. Kipp, J.-P. Martin, P. Paggio, and D. Heylen, 92-98.

[2] https://www.phonetik.uni-muenchen.de/Bas/BasSaGADoku.pdf

**Nominate For A Reproducibility Award:**

No code available.

---

### Decision · Program_Chairs · 2024-07-30

**Decision:**

Accept

**Comment:**

This paper investigates the reliability of gesture annotations in the BEAT dataset through three analyses: re-annotation of a segment, a word recognition test on lexical affiliates, and an examination of gesture durations. The results reveal systematic inaccuracies and outliers, prompting a discussion and suggestions on improving gesture-annotation methodologies.

Reviewer AD8U highlights the paper's significance and detailed discussions, suggesting the inclusion of concrete methodological examples and missing references like the SaGA dataset. Reviewer rK81 appreciates the paper’s organization and reproducibility but notes minor presentation issues and the lack of annotated gesture phases.

Both reviewers agree on the paper's importance and its potential contribution to the field, recommending acceptance. Ensure the camera-ready version addresses the highlighted weaknesses, including methodological examples and related work references.